# School Feeding as a Protective Factor against Insulin Resistance: The Study of Cardiovascular Risks in Adolescents (ERICA)

**DOI:** 10.3390/ijerph191710551

**Published:** 2022-08-24

**Authors:** Aline Bassetto Okamura, Vivian Siqueira Santos Gonçalves, Kênia Mara Baiocchi de Carvalho

**Affiliations:** Graduate Program in Public Health, Faculty of Health Sciences, University of Brasília, Campus Universitário Darcy Ribeiro S/N, Asa Norte, Brasília 70910-900, Brazil

**Keywords:** adolescent, insulin resistance, school feeding, adolescent health, school food environment, multilevel analysis

## Abstract

The objective of this study was to use ERICA data from adolescents from Brazilian public schools to investigate the role of school feeding in insulin resistance markers. Public school students (12–17 years old) with available biochemical examinations were selected. Adolescents answered a self-administered questionnaire, and contextual characteristics were obtained through interviews with principals. A multilevel mixed-effects generalized linear model was performed at the contextual and individual levels with each insulin resistance marker (fasting insulin, HOMA-IR, and blood glucose levels). A total of 27,990 adolescents were evaluated (50.2% female). The prevalence of (1) altered insulin was 12.2% (95% CI; 11.1, 13.5), (2) high HOMA-IR was 24.7% (95% CI; 22.8, 26.7), and (3) high blood glucose was 4.6% (95% CI; 3.8, 5.4). School feeding was positively associated with an insulin resistance marker, decreasing by 0.135 units of HOMA-IR (95% CI; −0.19, −0.08), 0.469 μU/L of insulin levels (95% CI; −0.66, −0.28), and 0.634 mg/dL of blood glucose (95% CI; −0.87, −0.39). In turn, buying food increased blood glucose by 0.455 mg/dL (95% CI; 0.16, 0.75). School feeding was positively associated with insulin resistance variables, demonstrating the potential of planned meals in the school environment to serve as a health promoter for the adolescent population.

## 1. Introduction

Adolescence, which includes persons aged 10–19, is a transitional period characterized by important biological, cognitive, emotional, and social changes. Additionally, this period is marked by an increase in autonomy and independence in relation to their families and a growing desire for new behavior and experiences. Behaviors created and established during this period can be carried into adulthood, affecting several health-related issues, such as unhealthy practices and chronic diseases [1].

Schools are an important social environment for adolescents, where peer interaction, emotional control, behaviors, and attitudes related to health and food consumption are promoted, which are able to determine their health in adult life [2,3]. In addition, school is an influencing environment, with an important role in health promotion, which can help reduce health inequities [4].

Nutrition programs and policies are implemented through a healthy school environment by offering adequate meals and making it a health-promoting place, thereby improving health and nutritional status and food consumption and reducing the risk of chronic diseases, such as obesity and hypertension [3,5]. In Brazil, the National School Feeding Program (PNAE) serves all public schools offering planned, healthy, and free meals, which must provide 70% of the nutritional needs of those who study full-time and 20% of those who study part-time; thus, this program is considered one of the most important public health programs in the world [6]. Some studies have investigated the relationship of specific policies in the school environment with effects on metabolic risk, eating behaviors, and adiposity [7], and have also shown that the school environment can be both protective and risky for cardiovascular disease [8]. However, few studies have directly addressed the school food environment, including planned school meal consumption, and the potential effects of insulin resistance markers in a representative sample of adolescents.

According to the American Diabetes Association, there are some methods for the assessment of insulin resistance, such as the homeostatic model assessment for insulin resistance (HOMA-IR), but screening for type 2 diabetes mellitus in individuals with altered blood glucose is also recommended [9]. In addition, early insulin resistance detection is important as part of the cardiovascular risk factors for metabolic syndrome (MS), which is responsible for the increase in cardiovascular disease and mortality [10]. According to data from the Study of Cardiovascular Risks of Adolescents (ERICA in Portuguese) [11], approximately 2.8% of Brazilian adolescents from public schools had MS, with insulin resistance present in 19% of the sample, evaluated by HOMA-IR [12]. Furthermore, the prevalence of overweight and obesity was 17% and 8.4%, respectively, and 14.4% presented prehypertension [11,13]. Altogether, these data point to the potential cardiovascular risk, which reinforces the importance of increasing attention to this population.

In this context, biochemical markers in addition to the prevalence of specific clinical conditions help to understand the health status of the adolescent population, which seems to be strongly associated with the food environment. Considering the dimensions of the country, the importance of adolescent health, and the magnitude of the PNAE, this study conducted unprecedented analysis using data from the ERICA from Brazilian public schools to investigate the role of school feeding in insulin resistance markers.

## 2. Materials and Methods

### 2.1. Study Design

This study analyzed national data from the ERICA, which is a cross-sectional, school-based study including school adolescents aged 12 to 17 years. The ERICA study aimed to estimate the frequency of cardiovascular risks in this population. The study was conducted between 2013 and 2014 and evaluated approximately 75,000 adolescents from public and private schools in urban and rural areas in 124 Brazilian municipalities. The students answered a self-administered questionnaire using the LG GM750Q personal digital assistant model about themselves or their family members. For the environmental data, the school principals participated in an interview in which a questionnaire about the school was completed by the interviewers.

### 2.2. Setting and Participants

The ERICA included students who lived in Brazilian municipalities with more than 100,000 inhabitants and excluded those with temporary or permanent physical or mental disabilities and pregnant adolescents. For the present study, public school students and students who had collected blood for biochemical examinations were selected.

The ERICA sample was stratified into 32 geographic strata constituting the 27 Brazilian capitals and 5 more macroregions. For each geographic stratum, schools were selected with a probability proportional to size and inversely proportional to the distance from the capital. The ERICA sample is representative of larger municipalities at the national, regional, and Brazilian capital levels. More details about the sample and study design can be found in Bloch et al. (2015) [14] and Vasconcellos et al. (2015) [15].

In the present study, data from 27,990 adolescents were analyzed, corresponding to a response rate of eligible individuals of 44.7% (Figure 1).

### 2.3. Variables and Categories

For the present analysis, the adolescents’ individual and contextual (environment) characteristics were considered. From the individual characteristics, the following variables were analyzed: age, sexual maturation, waist circumference, physical activity, mother’s education, ethnicity/skin color, self-reported consumption of meals prepared in school, purchased food at school, HOMA-IR, insulin, and blood glucose levels. For the environmental characteristics, the following information from the school’s questionnaire was considered: location area, sale of food at school, presence of vending machines, presence of advertising of industrialized foods, and sale of food in the vicinity of the school (Table 1).

### 2.4. Insulin Resistance Markers (Outcome)

Biochemical analysis was performed only with students who studied in the morning and who provided blood samples after obtaining written permission from their parents or guardians. Adolescents were instructed to fast for 12 h before the exam, and analyses of glucose and insulin were applied. Blood samples were processed and separated into plasma and serum within 2 h after collection and stored at temperatures ranging from 4 to 10 °C.

The hexokinase method was used for glucose analysis and chemiluminescence for insulin analysis. For HOMA-IR calculation, the formula described by Matthews et al. (1985) [16] was used: insulin (mU/L) × (glucose [mg/dl] × 0.0555)/22.5.

The insulin level cutoff points used for the assessment were desirable (<15 mU/L), borderline (15–20 mU/L), and high (≥20 mU/L) [17]. Borderline and high insulin levels were included in the undesirable group. The HOMA-IR cutoff points used were described by Chissini et al. (2020) [12] as follows: 2.80 for general adolescents, 2.32 for female adolescents, and 2.87 for male adolescents. The following blood glucose cutoff points used were recommended by the American Diabetes Association (2020) [9]: normal fasting blood glucose < 100 mg/dL and undesirable blood glucose ≥ 100 mg/dL.

### 2.5. Contextual Characteristics (Exposure Variables Related to School)

Schools were classified according to their location (rural or urban) and through the responses collected in the questionnaire or by observing the school environment. The following questions were asked: “Does the school offer meals prepared on your premises?” (yes or no); “At school, there are self-service machines that are working for the sale of food, such as soft drinks, sweets, potato chips, and others.” (yes or no); “Is there a way to sell food at school?” (yes or no); “What foods are sold? (sweets, candies, lollipops, chocolates, cookies, soft drinks, natural guarana, fried or baked snacks, sandwiches, pizzas, or others);” “Is there advertising for industrialized foods at school?” (yes or no); “What kind of advertising is there at school? (sweets, candies, lollipops, chocolates, cookies, soft drinks, natural guarana, fried or baked snacks, sandwiches, pizzas, or others);” “Is there a street vendor selling food or nonalcoholic drinks at the door or around the school?” (yes or no); “What is sold? (food, candies, chocolates, lollipops, popcorn, drinks, food, and drinks).”

### 2.6. Individual Characteristics (Exposure Variable Related to School)

Using the questionnaire answered by the adolescents, the answers to the following questions were used: “Do you eat the meal offered by the school?” (yes or no), and “Do you buy food in the school cafeteria?” (yes or no).

### 2.7. Individual Characteristics (Confounders)

The demographic characteristics analyzed were sex, age (<15 years and ≥15 years), ethnicity/skin color (white, black, or brown, Asian, indigenous, and does not know/prefers not to answer), and mother’s education (illiterate or elementary school incomplete, elementary or high school complete, complete or incomplete higher education, or does not know/does not remember). Sexual maturation was classified to Tanner’s stages [18], which assessed adolescents’ self-perception validated method, according to images of breasts for girls, genitalia for boys, and pubic hair for both, and classified as pubertal if the adolescent fulfilled at least one puberty characteristic (stage 4 or 5). Central obesity was defined based on the waist circumference criteria (percentile > 90: female ≥ 82.6 and male ≥ 86.2). Adolescents were in light clothes, without shoes, and maintained proper body posture. Anthropometric assessments were performed by trained investigators; those who reported engaging in physical activity ≥ 300 min/week were considered sufficiently active, and those who did not reach this value (<300 min/week) were considered insufficiently active [19].

### 2.8. Statistical Analysis

In the descriptive stage, the prevalence and its respective confidence intervals (95%) were calculated to characterize the sample and determine the distribution of the variables of interest. The natural weights of the sample design and the use of poststratification estimators were considered. In the analytical stage, to investigate the factors associated with insulin resistance, a multilevel mixed-effects generalized linear model was performed at the contextual and individual levels with each marker (HOMA-IR, blood glucose, and insulin) in sequential steps and using school as a second level. This method of analysis was applied considering that the school is a cluster and that it can influence some individual characteristics analyzed. Mixed models aggregate fixed and random effects in the same analysis, which is indicated in the context where students are grouped in schools, but it is intended to identify the variance of the effect of the school environment for the outcome variables [20,21]. For the first stage, the independent variables were analyzed individually for each outcome, and those with *p* < 0.20 were selected for the adjusted multivariate analysis. Then, the variables sex, sexual maturation, physical activity, and maternal education were used to adjust the analysis. Statistical significance was set at *p* < 0.05.

In both models (individual and contextual levels), the original weights were considered from the sample design. Descriptive analyses were performed using the command survey (“svy”) in the statistical program Stata version 16. The command “meglm” was used for multilevel linear regression.

### 2.9. Ethical Aspects

The original ERICA project was approved by the Ethical Committee of the Federal University of Rio de Janeiro in 2009 (protocol number 45/2008) and in the other 26 states and the Federal District. Approval was also obtained from all state and local departments of education in all schools that participated in the study. This study was conducted according to the principles of the Declaration of Helsinki. The students signed a consent form, and the parents of those students who collected a blood sample provided written consent as well.

## 3. Results

Most adolescents were 15 years of age or older and declared themselves black or brown. Most of them ate school meals, although they also bought food sold at school. Regarding insulin resistance markers, there was a prevalence of 12.2% (95% CI; 11.1, 13.5) in undesirable insulin, 24.7% (95% CI; 22.8, 26.7) in high HOMA-IR, and 4.6% (95% CI; 3.8, 5.4) in undesirable blood glucose. There were greater levels of fasting insulin and HOMA-IR in female adolescents, but the most remarkable results were observed for blood glucose levels in males (Table 2).

Regarding school environment (Table 3), most schools were in urban areas, less than half of the students studied in schools that sold food, but more than half had access to food sold in the school’s immediate vicinity. Students reported (52.3%; 95% CI; 46.6, 58.9) daily exposure to a variety of processed foods and beverages in and around the school.

School feeding was positively associated with the insulin resistance marker HOMA-IR, both in the crude analysis and in the adjusted analysis, where eating school food decreased HOMA-IR by 0.135 units (95% CI; −0.19, −0.08). Similarly, eating the meal offered by the school corroborated the findings for insulin and blood glucose levels, which decreased by 0.469 μU/L (95% CI; −0.66, −0.28) and 0.634 mg/dL (95% CI; −0.87, −0.39), respectively (*p* < 0.001). Finally, buying food at school was considered a negative association for increased blood glucose levels by 0.455 mg/dL (95% CI; 0.16, 0.75; *p* = 0.002) (Table 4).

## 4. Discussion

This study, through a robust methodology, demonstrated that the consumption of school feeding provided by Brazilian public schools has a protective effect on the variables related to insulin resistance, such as HOMA-IR, insulin, and blood glucose. In turn, buying food in school cafeterias showed a significant inverse association with blood glucose levels, which suggests that this practice is a risk factor for increasing levels of this metabolic marker.

All schools analyzed in this study were public schools, which are part of the PNAE. The PNAE, in addition to carrying out food and nutrition education actions, offers planned, healthy, and free meals during the school term for children and adolescents that cover their nutritional needs at this time [6]. However, this is not the adolescents’ only option to eat while in school since they can purchase food in school cafeteria self-service machines and from places in the school’s immediate vicinity, all of which are highly present. Therefore, the adolescents’ option of choosing not to consume school feeding should be considered, which can have a negative impact on their health.

In the present study, behavior was a determinant in the investigated associations, and the simple fact of having other foods offered in cafeterias at school or in the surroundings and vending machines and advertising was not associated with insulin resistance markers. The behavior, whether consciously or automatically, occurs within a social and environmental context and is influenced by several factors, such as at the individual, social, economic, community, and family levels [22,23]. Adolescents’ choices and behaviors are quite complex and can vary according to gender, age, and parental education [23,24]; these variables were controlled in the present study to avoid possible biases.

Gonçalves et al. (2019) [3] investigated the eating environment in schools participating in the ERICA study to identify individual and contextual factors associated with hypertension and obesity, and found that there was a 35% lower chance of obesity among students who received school feeding compared with those who were enrolled in schools that did not offer meals, in addition to being less likely to have high blood pressure when they ate meals. Another study by the ERICA (Recife—PE) aimed to identify schools that promote healthy eating and physical activity and their relationship with overweight, hypertension, insulin resistance, and hypercholesterolemia in adolescents; this study found that there was a higher prevalence of overweight adolescents in those schools with an unhealthy food environment. In addition, those schools had a higher prevalence of students with hypercholesterolemia [25]. Cardiovascular risk factors associated with insulin resistance and visceral obesity are emerging at an increasingly early stage, such as during childhood and adolescence, and are associated with a high probability of chronic diseases in adulthood [26].

The demand for health services during adolescence is low, and biochemical assessments, particularly blood glucose, HOMA-IR, and insulin levels, can be important early markers of diabetes [9,27]. According to data from the National Adolescent School-Based Health Survey (2015) [28], a cross-sectional study consisting of Brazilian ninth graders and adolescents aged 13 to 17 years, only 53.2% of public school students use these services. Importantly, health changes in adolescence can be perpetuated into adulthood, including the development of diseases [1]. In the present study, we verified the high prevalence of high HOMA-IR (24.7%; 95% CI; 22.8, 26.7) and insulin (12.2%; 95% CI; 11.1, 13.5), despite having a low prevalence of hyperglycemia (4.6%; 95% CI; 3.8, 5.4). Early recognition of these risks, adequate treatment, and lifestyle modifications have been associated with reductions in MS in adults [26].

The incidence of diabetes has increased worldwide, as well as associated diseases, such as increased cardiovascular risk (increasing hypertension and dyslipidemia) and obesity [9,29]. Diabetics can also develop microvascular complications, such as diabetic kidney disease, retinopathy, and neuropathy [30]. In addition, it is estimated that the annual cost of diagnosed diabetes in 2017 was $327 billion, generating costs for individuals and society [31]. Strategies for early risk identification of diabetes development and control are necessary to ensure it does not continue into adulthood.

Although the ERICA is a large school-based study with Brazilian representation, some methodological limitations should be considered. The cross-sectional study design does not allow for the inference of causality, and the environment characterization represents adolescents, not schools. Information bias may be present since the questionnaire was self-reported by adolescents, and blood collection was performed in the morning after an overnight fast, yet it is not possible to verify whether this request was followed. The lack of information about the frequency of consumption of school meals provided by the PNAE, in addition to the frequency of purchase of food at the school cafeteria, also limits our power of inference from the findings of the present study. Even so, the methodological rigor of the ERICA study is highlighted, with team training and standard laboratory analysis methods, in addition to robust statistical analyses, with data weighting and control of possible confounding variables. Further studies are needed to investigate possible changes that may have occurred in the Brazilian school environment in recent years, as well as the results of isolated intervention programs.

## 5. Conclusions

This study presents the association of school feeding consumption as a protective effect for insulin resistance variables and demonstrates the risk of purchasing food to increase blood glucose level. The importance of providing food and nutrition education in schools, in order to increase students’ knowledge is evident, and the need to implement policies to promote health in the school environment is clear, with the aim of strengthening adherence to the school feeding program.

## Figures and Tables

**Figure 1 ijerph-19-10551-f001:**
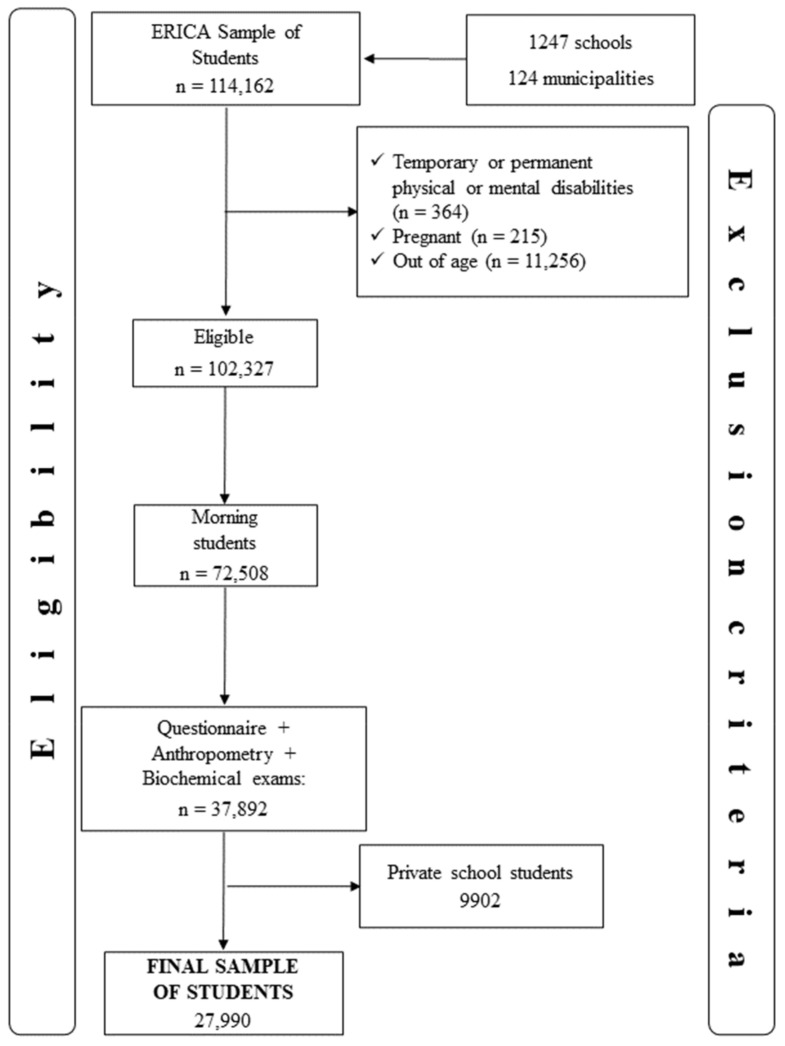
Flow chart of adolescents included. Study of Cardiovascular Risks in Adolescents (ERICA), Brazil, 2013–2014.

**Table 1 ijerph-19-10551-t001:** Individual and environmental characteristics variables. Study of Cardiovascular Risks in Adolescents (ERICA), Brazil, 2013–2014.

Individual Characteristics	Environmental Characteristics
Age (<15 years/≥15 years)	Location school area (Rural/Urban)
Sexual maturation (advanced-stage Tanner’s)	Sale of food at school (Yes/No)
Waist circumference (<percentile >90/> percentile >90)	Presence of vending machines (Yes/No)
Physical activity (sufficiently active/insufficiently active)	Presence of advertising of industrialized foods (Yes/No)
Mother’s education (illiterate or elementary school incomplete, elementary or high school complete, complete or incomplete higher education, or does not know/does not remember)	Sale of food in the vicinity of the school (Yes/No)
Ethnicity/skin color (white, black, or brown, Asian, indigenous, and does not know/prefers not to answer)	
Self-reported consumption of meals prepared in school (Yes/No)	
Purchased food at school (Yes/No)	
Blood glucose levels (Normal/Undesirable)	
Insulin levels (Borderline/High insulin)	
HOMA-IR (Normal/High)	

**Table 2 ijerph-19-10551-t002:** Adolescents’ general characteristics by sex. Study of Cardiovascular Risks in Adolescents (ERICA), Brazil, 2013–2014.

Individual Characteristic	Total (*n* = 27,990)		Male (*n* = 13,951)		Female (*n* = 14,039)	
	%	95% CI	%	95% CI	%	95% CI
Age ≥ 15 years	53.4	-	53.0	-	53.8	-
Advanced stage of sexual maturation ^a^	75.8	74.7–76.9	78.1	77.1–79.3	73.6	72.1–75.2
Ethnicity/skin color						
White	35.4	33.0–37.9	35.4	33.1–38.3	35.4	33.1–38.0
Black or brown	59.7	57.6–61.8	58.5	56.0–63.2	61.0	59.0–63.2
Asian	1.9	1.5–2.3	1.9	1.4–2.5	1.9	1.5–2.3
Indigenous	0.7	0.5–0.9	0.9	0.7–1.4	0.4	0.3–0.5
Does not know/prefers not to answer	2.4	2.0–2.8	3.3	2.7–4.0	1.4	1.1–1.8
Mother’s education						
Illiterate or ES incomplete	23.0	20.5–25.8	23.1	19.5–27.0	23.0	21.2–25.0
ES or HS complete	33.5	31.4–35.6	32.1	29.3–35.0	35.1	33.1–37.0
Complete or incomplete HE	14.2	12.8–15.7	15.2	13.3–17.3	13.2	11.7–15.1
Does not know/does not remember	29.3	27.8–30.8	30.1	27.6–32.0	29.1	27.3–30.6
Consumption of school feeding	62.4	59.9–64.8	64.4	61.2–68.0	60.3	57.4–63.2
Purchase of foods at the school cafeteria	62.8	57.9–67.4	64.7	59.4–69.5	60.9	56.1–65.5
Physically active ^b^	52.4	51.2–53.5	63.8	62.2–65.4	41.1	39.3–43.1
Central obesity (waist circumference ^c^)	11.2	10.2–12.3	11.2	9.9–12.6	11.2	9.9–12.6
Borderline or high insulin ^e^	12.2	11.1–13.5	9.7	8.4–11.1	14.8	13.3–16.4
High HOMA-IR ^d^	24.7	22.8–26.7	14.5	12.9–16.4	35.1	32.1–37.5
Undesirable blood glucose ^f^	4.6	3.8–5.4	6.2	4.9–7.7	3.1	2.4–3.7

ES: elementary school; HS: high school; HE: higher education; HOMA-IR: homeostasis model assessment; CI: confidence interval. ^a^ Advanced stage of sexual maturation according to Tanner’s criteria (stage 4 or 5); ^b^ sufficiently active ≥ 300 min/week, insufficiently active < 300 min/week; ^c^ percentile value > 90: female ≥ 82.6 cm and male ≥ 86.2 cm; ^d^ female > 2.32 and male > 2.87 [12]; ^e^ desirable <15 mU/L, borderline or high ≥15 mU/L [18]; ^f^ normal <100 mg/dL; and undesirable ≥100 mg/dL [9].

**Table 3 ijerph-19-10551-t003:** Characteristics of school environment. Study of Cardiovascular Risks in Adolescents (ERICA), Brazil, 2013–2014.

Characteristic	Public Schools (% Students)	
	%	95% CI
School location area		
Rural area	6.1	1.8–19.1
Urban area	93.9	80.9–98.2
School foods environment		
Sale of food at school	42.9	35.9–50.3
Sweets, candies, lollipops, chocolates, etc.	35.0	28.5–42.1
Sweet or salty cookies	30.7	25.3–36.6
Soft drinks	32.4	27.1–38.3
Natural guarana	8.9	6.7–11.8
Snacks (fried or baked)	38.3	31.5–45.6
Sandwiches and/or pizza	27.9	22.0–34.7
Sale of food in school vending machines	5.2	3.3–7.9
Food	1.1	0.6–1.7
Drinks	3.6	1.9–6.5
Food and drinks	0.5	0.2–1.3
Advertisement of industrialized foods at school	3.0	1.8–4.8
Sweets, candies, lollipops, chocolates, etc.	1.3	0.8–2.2
Sweet or salty cookies	1.0	0.2–3.3
Soft drinks	1.7	1.1–2.8
Sandwiches and/or pizza	0.08	0.02–0.3
Noncarbonated sugary drinks (mate, tea, isotonic, natural guarana)	0.7	0.3–1.6
Others	1.1	0.4–3.3
Sale of food in the school’s immediate vicinity	52.3	46.6–58.9
Food	28.6	23.6–34.3
Drinks	3.9	2.2–7.0
Food and drinks	19.8	15.6–24.7

CI: confidence interval.

**Table 4 ijerph-19-10551-t004:** Multilevel linear regression of contextual and individual level and HOMA-IR, insulin, and glucose markers. Study of Cardiovascular Risks in Adolescents (ERICA), Brazil, 2013–2014.

Characteristics	Crude Analysis			Adjusted Analysis ^1^		
	Coef.	95% CI	*p*-Value	Coef.	95% CI	*p*-Value
**HOMA-IR**						
**Contextual level**						
Sale of foods at school	0.001	−0.08; 0.08	0.970			
Sale of food in school vending machines	−0.037	−0.21; 0.12	0.649			
Advertisement of industrialized foods at school	−0.097	−0.28; 0.09	0.300			
Sale of food in the school’s immediate vicinity	−0.051	−0.13; 0.03	0.210			
**Individual level**						
Consumption of school feeding	−0.141	−0.19; −0.09	<0.0001	−0.135	−0.19; −0.08	<0.0001
Purchase of foods at the school cafeteria	0.022	−0.03; 0.08	0.447			
**INSULIN**						
**Contextual level**						
Sale of foods at school	0.007	−0.32; 0.33	0.967			
Sale of food in school vending machines	−0.200	−0.87; 0.47	0.556			
Advertisement of industrialized foods at school	−0.301	−1.06; 0.46	0.441			
Sale of food in the school’s immediate vicinity	−0.106	−0.43; 0.22	0.521			
**Individual level**						
Consumption of school feeding	−0.503	−0.68; −0.32	<0.0001	−0.469	−0.66; −0.28	<0.0001
Purchase of foods at the school cafeteria	0.025	−0.18; 0.24	0.813			
**GLUCOSE**						
**Contextual level**						
Sale of foods at school	−0.097	−0.58; 0.39	0.695			
Sale of food in school vending machines	0.171	−0.83; 1.17	0.737			
Advertisement of industrialized foods at school	−0.637	−1.78; 0.51	0.274			
Sale of food in the school’s immediate vicinity	−0.429	−0.92; 0.06	0.084			
**Individual level**						
Consumption of school feeding	−0.630	−0.87; −0.39	<0.0001	−0.634	−0.88; −0.39	<0.0001
Purchase of foods at the school cafeteria	0.613	0.33; 0.89	<0.0001	0.455	0.16; 0.75	0.002

CI: confidence interval. ^1^ Model were adjusted for sex, sexual maturation, physical activity, and maternal education.

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
