# Peer review of "School Feeding as a Protective Factor against Insulin Resistance: The Study of Cardiovascular Risks in Adolescents (ERICA)"

_ijerph, 2022, doi:10.3390/ijerph191710551_

Round 1

Reviewer 1 Report

In introduction section, the paper should add research background and research importance. The paper focuses the effect of school feeding on adolescents health and use disease prevalence to measure individual health. There is no information giving a specific age range to define adolescents.

In data and methods section, variable definitions may be more clear by listed in a table. For methods the paper should add the reason why use a multilevel mixed-effect generalized linear model.

It would be better to divide result into sample descriptive analysis and empirical analysis.

In conclusion section,  strategies need close to the conclusion such as increasing dietary nutrition knowledge of adolescents.

The range of explained variable “insulin resistance markers” was confusing. Line 94-95 refer to” insulin resistance markers, including HOMA-IR, insulin, and blood glucose levels.” Section 2.4 also show that insulin resistance contained HOMA-IR, insulin, and blood glucose levels.

However, line 250 mentioned “particularly blood glucose levels and insulin resistance”

Some problem about the variable “consumption of school feeding”

First, table 2 show that 62.4% adolescents consumption of school feeding and 62.8% adolescents purchase of foods at the school cafeteria in the sample. So we can see there was a group of people have eaten in both places. This question only illustrated the individual have ever consumption of school feeding or purchase of foods at the school cafeteria without information about frequency. The effect of diet on chronic diseases occurs in the long term. The question should be changed to ask about the frequency of personal dining places or high-frequency dining places.

Additionally, a misspelling of Table 2 is” consuption of school feeding”.

Reviewer 2 Report

Thank you very much for allowing me to review the article entitled “School feeding as a protective factor against insulin resistance: the Study of Cardiovascular Risks in Adolescents (ERICA)” (ijerph-1763394).

This article is presented for the special Issue “Healthy Lifestyles for the Prevention of Non-communicable Diseases: Epidemiological Findings and Interventions” within the section “Health Behavior, Chronic Disease and Health Promotion”.

“The aim of this study conducted unprecedented analysis using data from the ERICA from Brazilian public schools to investigate the protective role of school feeding on insulin resistance markers.”

It is a very powerful study in terms of sample size. For a scientific journal I think the objective should be more neutral, I suggest something similar to this:

The objective of this study was, using ERICA data from adolescents from Brazilian public schools, to investigate the role of school feeding in markers of insulin resistance.

In the summary, the characteristics of the adolescents should be indicated, for example, what ages are studied, the concept of adolescent is very variable in the studies, I recommend that it be specified. The concept "insulin resistance marker" must be indicated on what it is based.

The introduction should be expanded with previous studies on this subject that serve to introduce the importance of the study in Brazil.

material and method.

It is well described, but it should be noted that it is a study carried out between 2013 and 2014, the response rate should be added.

I recommend reviewing the manuscript for editing.

Results: I suggest that the diagram of selection of the participants in the study I passed to material and methods.

The results should be more widely described.

Discussion: I suggest the elimination of adjectives, it must be remembered that this is a scientific journal, therefore these adjectives reduce the quality of the study.

it should be expanded and compared with other studies on this topic.

Round 2

Reviewer 1 Report

The wiring is more fluid than before and the paper points the limitation about the study.Overall, the introduction could be more to the point.

Reviewer 2 Report

After carefully reviewing the article “School feeding as a protective factor against insulin resistance: the Study of Cardiovascular Risks in Adolescents (ERICA)” V2 (ijerph-1763394) and the comments of the authors to the suggestion made on it.

I considered that it is an interesting article that allows us to know the role of school feeding in markers of insulin resistance in the youth population (12 to 17 years old) of Brazil in the period 2013-14. It is a study in which the strength of the sample size stands out and it presents as an important weakness the cross-sectional design and the period in which it was carried out, but as the authors indicate, there is no updated information on this subject, and they considered that it should be published.